# Phenyllactic Acid as a Marker of Antibiotic-Induced Metabolic Activity of Nosocomial Strains of *Klebsiella pneumoniae* In Vitro Experiment

**DOI:** 10.3390/microorganisms13112599

**Published:** 2025-11-15

**Authors:** Maria Getsina, Ekaterina Chernevskaya, Ekaterina Sorokina, Tatiana Chernenkaya, Natalia Beloborodova

**Affiliations:** 1Federal Research and Clinical Center of Intensive Care Medicine and Rehabilitology, 25-2 Petrovka Str., 107031 Moscow, Russia; echernevskaya@fnkcrr.ru (E.C.);; 2State Budgetary Healthcare Institution N.V. Sklifosovsky Research Institute of Emergency Care of the Moscow Health Department, 129090 Moscow, Russia

**Keywords:** *Klebsiella pneumoniae*, multidrug-resistant strains, hospital infection, antibiotics, metabolic activity of bacteria, phenyllactic acid, GC-MS, microbial metabolites

## Abstract

*Klebsiella pneumoniae* (*K. pneumoniae*) is a major nosocomial pathogen with increasing antibiotic resistance. Treatment failures and high mortality rates in sepsis caused by *K. pneumoniae* are associated with difficulties in choosing an adequate antibacterial therapy in the presence of resistance to all available antibiotics, based on the results of susceptibility tests. This study aimed to identify “weak points” in the metabolism of *K. pneumoniae*, to be able to use these features in the future. Ten nosocomial *K. pneumoniae* strains were incubated with fourteen broad-spectrum antibiotics representing major drug classes. Aromatic metabolites were analyzed using gas chromatography–mass spectrometry after 24 h exposure. Phenyllactic acid (PhLA), comprising 86% of detected phenylcarboxylic acids, served as the metabolic activity marker. Antibiotics demonstrated multidirectional effects on aromatic compound metabolism. Doxycycline, nitrofurantoin, rifampicin, and tigecycline significantly suppressed metabolic activity, confirmed by decreased PhLA levels. Conversely, meropenem, cephalosporins (ceftriaxone, cefepime, cefotaxime, and ceftazidime), ciprofloxacin, and amikacin stimulated PhLA production, suggesting that bacterial metabolic activity was maintained despite the presence of antibiotics. PhLA is a promising biomarker for quantifying *K. pneumoniae’s* metabolic response to antibiotics. This potentially introduces a novel approach for future investigations into resistance mechanisms and has the potential to increase the effectiveness of therapies for multidrug-resistant *K. pneumoniae* infections by providing an additional analytical tool to traditional susceptibility testing methodologies.

## 1. Introduction

*Klebsiella pneumoniae* (*K. pneumoniae*) is one of the leading pathogens causing infectious diseases in intensive care units, due to its high virulence and ability to form resistant hospital strains [1]. The infection caused by *K. pneumoniae* can involve nearly all organs and tissues, with the respiratory tract, urinary system, and purulent-septic complications being the most commonly affected [2]. This pathogen is responsible for approximately 11.8% of all nosocomial pneumonias worldwide [3]. In case of generalized infection, bacteremia and sepsis may develop, which cause high morbidity and mortality [4,5].

*K. pneumoniae*, included in the group of especially dangerous infections (ESKAPE) [6], is characterized by high adaptability and the ability to develop resistance to modern antimicrobial drugs, including carbapenems, colistin, and tigecycline [7].

Improving diagnostic, preventive, and therapeutic strategies for infections caused by *K. pneumoniae* remains one of the priority tasks in modern medicine. Current clinical management faces significant challenges: timely and accurate pathogen identification; selection of effective antimicrobial therapy based on susceptibility profiles; development of novel treatment approaches for resistant strains.

The study of bacterial metabolic profiles provides promising avenues for developing novel diagnostic methods and therapeutic strategies against carbapenem-resistant strains [8]. This approach is particularly relevant for *K. pneumoniae*, whose metabolic profile demonstrates remarkable diversity [9,10,11,12]. Clinical isolates of *K. pneumoniae* exhibit significantly elevated metabolite production compared to control strains [13]. Notably, antibiotic-resistant variants display distinctive metabolic signatures. For instance, exposure to meropenem induces divergent metabolic responses: in wild-type strains, altered methane and amino acid metabolism; in KPC-producing strains, modified pyruvate metabolism [14]. Alterations in the bacterial metabolome are closely intertwined with antimicrobial resistance mechanisms, offering significant diagnostic potential. Metabolomic data help guide timely and appropriate antimicrobial therapy, optimizing treatment strategies [13]. Additionally, specific metabolic signatures show promise as potential biomarkers for the detection of bacteremia, expanding diagnostic capabilities in clinical microbiology [15].

Clinical observations reveal a marked accumulation of specific aromatic microbial metabolites in sepsis cases irrespective of etiology, including phenyllactic acid (PhLA), *p*-hydroxyphenyllactic acid, and *p*-hydroxyphenylacetic acid [16]. Under physiological conditions, aromatic metabolite biosynthesis primarily occurs in the intestinal environment, where microbial communities process intermediate compounds, with only minimal amounts of final metabolites entering systemic circulation [17,18]. Experimental evidence confirms a statistically significant elevation in PhLA concentrations within gut pathobiota samples obtained from sepsis patients, when compared to normobiota controls (significance level *p* = 0.002) [16].

The relationship between bacterial metabolic state and antimicrobial resistance is particularly noteworthy. Antibiotic efficacy frequently depends on the inhibition of central metabolic or biosynthetic pathways. When these pathways are disrupted, metabolic flux through aromatic amino acid metabolism—the pathway responsible for PhLA synthesis—decreases. Conversely, resistant bacterial strains maintain energy homeostasis and redox balance, frequently sustaining or even enhancing the production of key metabolites such as PhLA [19,20,21].

Studying bacterial metabolic profiles may help to develop new diagnostic methods and antimicrobial strategies to combat the global crisis against carbapenemase-positive *K. pneumoniae* [14].

The aim of the study was to investigate antibiotic-induced changes in the production of aromatic metabolites among nosocomial strains of *K. pneumoniae* in vitro.

## 2. Materials and Methods

### 2.1. General Description of the Experiment—Preparation of Microbial Suspension

The experimental design is shown in Figure 1. To begin the experiment, hospital strains of *K. pneumoniae* were isolated. The antibiotic-impregnated disk was placed in the medium together with a suspension of a daily culture of *K. pneumoniae* and incubated at 37 °C for 24 h; after that, the concentration of metabolites in the resulting suspension was determined using the gas chromatography–mass spectrometry method.

### 2.2. Preparation of Microbial Suspension: Obtaining Clinical Material, Primary Culture, Isolation, Identification and Determination of Sensitivity of Microorganisms

Samples of biomaterial for obtaining pure Klebsiella cultures for use in the experiment were obtained from the N.V. Sklifosovsky Research Institute of Emergency Care in a certified clinical microbiology laboratory in 2024. Hospital strains of *K. pneumoniae* were obtained during routine examination of patients and isolated from various types of clinical material (Table 1). To anonymize the material, hospital strains and hospital numbers were replaced by letter designations. Primary culture of the clinical material was performed using conventional methods.

#### 2.2.1. Blood Culture

Blood was collected from patients via a peripheral vein using aseptic technique. For each test, 10 mL of blood was collected simultaneously into two commercial vials for the JUNONA LABSTAR blood culture analyzer (Scenker Biological Technology Co., Ltd., East End of Wei’er Road, Fenghuang Industrial Zone, Liaocheng, Shandong Province, China) containing commercial culture media:

JUNONA^®^ Nutrient Medium with Antibiotic Neutralizers for Cultivation of Anaerobes (Scenker Biological Technology Co., Ltd, East End of Wei’er Road, Fenghuang Industrial Zone, Liaocheng, Shandong Province, China).

The resulting samples were delivered to the laboratory and placed in the Unona Labstar 100 blood culture analyzer (Scencer, Shanghai, China). If microbial growth was detected, a Gram-stained smear from the vial contents was examined microscopically. The vial contents were then plated on solid culture media: Schadler agar, 5% sheep blood agar, mannitol salt agar, Endo medium, and Sabouraud medium, and incubated in an incubator at 35 °C for 24–48 h.

#### 2.2.2. Culture of Fluid Obtained During Bronchoalveolar Lavage

Clinical specimens were collected during bronchoscopy. The obtained fluid was delivered to the laboratory within 2 h of collection. Initial culture of the delivered samples was performed on solid nutrient media: 5% sheep blood agar, chocolate and mannitol-salt agar, Sabouraud and Uriselect media, followed by incubation in an incubator at 35 °C for 24–48 h.

#### 2.2.3. Culture of Wound Secretions, Punctures, and Biopsies

Sample collection was performed in accordance with Methodological Guidelines MU 4.2.203905 “Technique for collecting and transporting biomaterial to microbiology laboratories.” Commercial tubes with Amies medium were used for the storage and transportation of the collected material. In the microbiology laboratory, the delivered material was cultured on Petri dishes with solid nutrient media (5% blood agar, mannitol-salt agar, Endo, Sabouraud, and Schadler media) and in semi-solid thioglycollate medium, which were incubated for 24–48 h in an incubator at 35 °C.

### 2.3. Species Identification of Microorganisms Isolated from All Types of Clinical Material

Species identification of microorganisms isolated from all types of clinical material was performed using a Vitek MS mass spectrometer (bioMerieux, Marcy-l’Étoile, France).

Antibiotic susceptibility testing of the isolated bacteria was performed using a Vitek 2 Compact automated microbiological analyzer (BioMerieux, France) or by the disk diffusion method on Mueller–Hinton agar using paper disks (Bioanalyse, Yenimahalle, Turkey; BD, Franklin Lakes, NJ, USA). Susceptibility testing results for the isolated pathogens were interpreted based on the current version of the EUCAST criteria. The results of determining the sensitivity of the isolated pathogens were interpreted based on the current version of the EUCAST criteria (Appendix A). It is evident that imipenem, cefepime, and meropenem showed resistance for all strains isolated for the experiment (Appendix A). In addition, the isolated strains showed polyresistance to a wide range of antibiotics. Individual strains showed sensitivity to amikacin (3 out of 10, 30% of strains), tigecycline (1 out of 3, 33%), colistin (3 out of 4, 75%), and tobramycin (1 out of 3, 33%).

### 2.4. Experiment with the Obtained Cultures and the Addition of Antibiotics

A suspension was prepared in a sterile saline solution from a daily culture of selected *K. pneumoniae* strains with a microbial concentration in accordance with the turbidity standard of 0.5 McFarland (microbial cell concentration 1.5 × 10^8^ cells/mL). Eppendorf tubes were filled with 1 mL of thioglycollate medium, 50 µL of the prepared suspension of a 24 h *K. pneumoniae* culture, and one of the standard antibiotic susceptibility testing disks. Commercial antibiotic susceptibility testing disks from BD (USA) and Bioanalyse (Turkey) were used for this study. Commercially produced disks contain the antibiotic concentration recommended for susceptibility testing according to EUCAST standards. The antibiotic concentrations used are listed in Table 2. Tubes containing the following served as controls:1 mL of thioglycollate medium;1 mL of thioglycollate medium and 50 µL of *K. pneumoniae* microbial suspension;1 mL of thioglycollate medium and a meropenem disk;1 mL of thioglycollate medium and a doxycycline disk;1 mL of thioglycollate medium and an amikacin disk.

The following antibiotics were used: nitrofurantoin, doxycycline, rifampicin, clarithromycin, meropenem, imipenem, cefepime, cefotaxime, ceftriaxone, ceftazidime, ciprofloxacin, tigecycline, amikacin, and cotrimoxazole. All the tubes were incubated in a thermostat at 37 °C for 24 h, then further analysis was carried out using the GC-MS method.

### 2.5. PCR Analysis of Microorganisms After an Experiment with Antibiotics

Quantitative assessment of the presence/growth of *K. pneumoniae* after incubation in suspension was analyzed using AmpliPrime “FLOROSCREEN-Aerobes” reagent kit (NextBio, Moscow, Russia), which includes reagents for DNA extraction, specific PCR primers specific for the determination of DNA from enterobacteria (Enterobacteriaceae family, including Klebsiella spp. in biological material by real-time PCR with hybridization-fluorescence detection on a multichannel platform compatible with CFX systems (Bio-Rad, Hercules, CA, USA). The kit description and intended use according to the manufacturer’s documentation for AmpliPrime “FLOROSCREEN-Aerobes”, including quantitative calibration via Ct-based standard curves and internal controls for process monitoring, are demonstrated in Appendix A. DNA was extracted by sequentially adding 10 μL internal control (Internal control sample for PCR with electrophoretic detection) and 20 μL universal sorbent into sterile tubes, followed by 300 μL lysis buffer and 100 μL clinical sample. Samples were vortexed, incubated at 65 °C, washed twice, dried at 65 °C, and eluted with 100 μL DNA elution buffer. After final centrifugation, supernatants containing purified DNA were collected for PCR. DNA samples were stored at 2–8 °C for up to one week of preservation.

### 2.6. Analysis of Metabolites Using a Gas Chromatograph Mass Spectrometer

To isolate metabolites from the suspension, double extraction with diethyl ether was performed according to the standard method described in [22]. Internal standard (100 μL) and distilled water (700 μL) were added to an aliquot of the microbial suspension (200 μL). For more accurate quantitative analysis using liquid–liquid extraction, surrogate internal standards were used that were absent in the studied samples: D5-benzoic acid for the conversion of benzoic acid; and 3,4-dihydroxybenzoic acid for phenylpropionic, phenylacetic, phenyllactic, *p*-hydroxyphenylpropionic, homovanillic, *p*-hydroxybenzoic, *p*-hydroxyphenylacetic acids; and D3-hydroxyphenyllactic acid for *p*-hydroxyphenyllactic acid. Derivatization to obtain trimethylsilyl derivatives was carried out under the following conditions: 20 μL of N, O-bis(trimethylsilyl)trifluoroacetamide was added to the resulting dry residue and kept at 80 °C for 15 min. The solution with trimethylsilyl derivatives was cooled at 5 °C for 30 min and diluted with 400 μL of n-hexane; 2 μL of the resulting solution was introduced into the GC–MS system. Metabolite analysis was performed on a SHIMADZU GCMS-QP2020 gas chromatograph mass spectrometer (Kyoto, Japan) using an SH-5ms capillary column (stationary phase—95% dimethylpolysiloxane—5% diphenylpolysiloxane, length—30 m, internal diameter—0.25 mm, stationary phase thickness—0.25 μm). Gas chromatographic separation conditions: injector temperature 260 °C, carrier gas (helium) flow rate 1.5 mL/min, flow split mode (1:5). Column thermostat temperature programming mode: initial temperature 80 °C for 4 min; then heating at a rate of 10 °C/min to 250 °C, holding at this temperature for 4 min. Total analysis time is 27 min. Mass spectrometric analysis conditions: electron ionization, electron energy 70 eV, interface temperature 250 °C, ionization chamber temperature 200 °C, *m*/*z* scanning range 50–450 amu, scanning rate three scans per second. Quantitative determination of metabolites was performed in the concentration range 0.4–50 μmol/L using the calibration dependence identified after analyzing the calibration solutions.

### 2.7. Calculation of the Metabolic Activity Coefficient

The coefficient of metabolic activity of the strain in relation to antibiotics (Kma) is calculated according to the equation:Kma (%) = (C_PhLA AB_ − C_PhLA control_)/C_PhLA control_) × 100,(1)
where C_PhLA AB_ is the concentration of PhLA in the suspension with the antibiotic and the strain under study, C_PhLA control_ is the concentration of PhLA in the control suspension with the strain under study.

### 2.8. Statistical Data Processing

The data were presented as medians (Mes) and interquartile ranges (IQR). All statistical calculations were performed using IBM SPSS Statistics v.27.0.

## 3. Results

### 3.1. Determination of Metabolites in the Substrate After Incubation of K. pneumoniae

Metabolic activity was studied using *K. pneumoniae* as an example during incubation in the presence of antibiotics. Ten hospital strains collected from different sites were used: bronchoalveolar lavage, whole blood, abscess contents, wound or drainage (Table 1). After 24 h of *K. pneumoniae* incubation in TGM, the concentration of nine microbial metabolites in the substrate, belonging to the class of phenylcarboxylic acids, was determined by GC-MS: benzoic acid (BA), phenylpropanoic acid (PhPA), phenylacetic acid (PhAA), phenyllactic acid (PhLA), *p*-hydroxybenzoic acid (*p*-HBA), p-hydroxyphenylacetic acid (*p*-HPhAA), *p*-hydroxyphenylpropanoic acid (*p*-HPhPA), homovanillic (HVA) and *p*-hydroxyphenyllactic acid (*p*-HPhLA). For 10 strains, the median concentrations of each acid were calculated and is shown in Table 3. Metabolites obtained in control tubes are also presented in Table 3, which were considered as background values during data analysis. Benzoic and phenylacetic acids were detected in control tubes containing only the antibiotic disk, as well as in the tube containing thioglycollate medium without additives, and their values remained virtually unchanged across all tubes.

The median concentrations of each acid taken for the metabolite values of the test tube corresponding to *K. pneumonia* without the addition of antibiotics (*n* = 10) were presented as percentages and combined in a diagram (Figure 2).

The main aromatic metabolite of *K. pneumoniae* is PhLA; its share is 86% among the phenylcarboxylic acids determined by GC-MS. The contribution of phenylpropanoic, phenylacetic, and *p*-hydroxyphenylpropanoic acids is less than 1% in total, and for the rest: benzoic, *p*-hydroxybenzoic, *p*-hydroxyphenylacetic, and *p*-hydroxyphenyllactic acids do not exceed 5% for each. Thus, PhLA was considered a potential marker of metabolic activity.

### 3.2. Metabolic Activity of K. pneumoniae Based on Changes in the Concentration of PhLA when Exposed to Different Classes of Antibiotics

Fourteen broad-spectrum antibiotics, commonly used in clinical practice, were used in the experiment to test their activity against most aerobic and anaerobic Gram-positive and Gram-negative bacteria.

The change in PhLA production after incubation in the presence of different antibiotics is shown for 10 hospital strains of *K. pneumoniae* (Figure 3).

The frequency of PhLA values exceeding the given antibiotic on the given strain, compared to the PhLA concentration at the control point where the antibiotic was not added, is shown in red. The PhLA concentration that is not too far from the PhLA value at the control point (±20%) is shown in yellow. The samples in which the PhLA concentration is lower than in the control are shown in green. The presence of disks with antibiotics such as doxycycline, nitrofurantoin, and rifampicin significantly suppresses the metabolic activity of the strain for most of the studied strains, which leads to a decrease in the PhLA concentration in the substrate. However, in the presence of disks with meropenem, ciprofloxacin, ceftriaxone, cefepime, cefotaxime, cotrimoxazole, and ceftazidime, the concentration of PhLA in the suspension in most cases exceeds or is close to the value in the control. This indicates the strain’s metabolic activity and its ability to produce metabolites despite the presence of the antibiotic. Clarithromycin and amikacin demonstrated a full spectrum of metabolic activity changes, from suppression to activation. In some cases, these data are consistent with the results of antibiotic susceptibility testing; more detailed experimental results are presented in Appendix A. It can be concluded that in the presence of resistance, the antibiotic fails to inhibit bacterial growth, which is accompanied by continued metabolic activity—specifically, increased PhLA concentrations.

### 3.3. Quantitative Assessment of Microbial Cell Numbers and Phenyllactic Acid Levels

An additional study using real-time PCR was conducted for four strains to determine the presence of *K. pneumoniae* in suspensions after incubation with antibiotics. Strains of various etiologies were selected, obtained from abscess contents, whole blood (two strains), and bronchoalveolar lavage. The results, combined with the concentration of PhLA in these suspensions, are presented in Table 4.

Since the PCR method reflects the content of microorganism DNA regardless of its presence in a living or non-living state, we almost never observed a decrease in the concentration of *K. pneumoniae* after incubation. This is due to the features of the PCR method. However, in the case of rifampicin, an increase in the concentration of *K. pneumoniae* was observed for all strains, while metabolic activity (PhLA production) was significantly reduced. For strains G, H, a similar picture was observed in the presence of doxycycline and nitrofurantoin, for strain I in the presence of nitrofurantoin and imipenem, and for strain J in the presence of amikacin, doxycycline, and nitrofurantoin. Cases when the concentration of PhLA increases simultaneously with the increase in the concentration of *K. pneumoniae* were observed for strain G in the presence of amikacin and meropenem, and for strain J in the presence of ceftazidime and clarithromycin.

### 3.4. Evaluation of Antibiotic Effectiveness for a Strain by the Coefficient Construction of a Metabolic Activity

The metabolic activity coefficient calculation is proposed as a tool for convenient visual analysis of the strain’s metabolic activity. Since the absolute values of the PhLA concentration may differ for different strains, all values are recalculated taking into account the value of the metabolite concentration in the medium without an antibiotic relative to the metabolite concentration with an antibiotic (Equation (1)). This allows us to answer the question about the production of metabolites, as well as to appreciate whether the strain is resistant or sensitive to these antibiotics. The peculiarity of the proposed method is that it is aimed primarily at assessing the metabolic activity of the microorganism strain under the influence of antibiotics.

The obtained quantitative data allow us to assess the metabolic activity (Figure 4) and select an antimicrobial drug for the patient that has the greatest effect on the metabolism of the microorganism.

From Figure 4, it is evident that the metabolic activity coefficient shows that all strains have low metabolic activity and, therefore, sensitivity to antibiotics such as doxycycline, nitrofurantoin, rifampicin (for all except strain G), as well as tigecycline (also except strain G) and imipenem (for all except strain H). The metabolic activity coefficient also shows that all strains have low metabolic activity and, therefore, sensitivity to antibiotics such as doxycycline, nitrofurantoin, rifampicin (for all except strain G), as well as tigecycline (also except strain G) and imipenem (for all except strain H). Clarithromycin, trimethoprim, meropenem, 3rd and 4th generation cephalosporins, ciprofloxacin, and amikacin, in most cases, exhibit high metabolic activity, potentially indicating low efficacy of the drug.

## 4. Discussion

This study, which examines changes in the metabolic activity of *K. pneumoniae* under the influence of various classes of antibiotics, is of significant scientific interest to a wide range of specialists, from fundamental microbiology to clinical practice and antimicrobial therapy. The study is particularly relevant because it not only provides a deeper understanding of the mechanisms of bacterial interaction with antimicrobials but also opens up new prospects for the development of innovative approaches to the diagnosis and treatment of infections caused by this pathogen.

### 4.1. Metabolic Profile of Klebsiella pneumoniae

The metabolic profile of *K. pneumoniae* was studied in detail using ten hospital strains isolated from various clinical specimens. A key finding was the predominance of phenyllactic acid (PhLA) in the metabolic profile, accounting for 86% of all detected phenylcarboxylic acids. Other metabolites, including phenylpropanoic acid, phenylacetic acid, and *p*-hydroxyphenylpropanoic acid, were found in significantly lower concentrations.

This is consistent with data showing that PhLA is known not as a metabolic product of humans, but also, above all, of a number of microorganisms [23]. In clinical medicine, PhLA is classified as a sepsis-associated aromatic microbial metabolite. Elevated PhLA levels in the serum of patients with sepsis and septic shock are an unfavorable prognostic factor [24]. Moreover, in vitro studies have shown that healthy microbiota are capable of biotransforming excess sepsis-associated metabolites, whereas in patients with sepsis, this function is impaired, leading to accumulation of metabolites in biological fluids [16].

Notably, PhLA functions as a natural antimicrobial agent produced by many bacterial strains [23]. Numerous studies demonstrate the effectiveness of PhLA against various pathogenic microorganisms, such as the periodontal pathogen *Aggregatibacter actinomycetemcomitans* [25], resistant strains of *K. pneumoniae* [24], biofilm forms of *Klebsiella oxytoca* [26], foodborne pathogens *Enterococcus faecalis* [27], *Listeria monocytogenes*, and *Escherichia coli* [28]. The antimicrobial properties of PhLA determine its wide practical application. In particular, it is actively used in the food industry to extend the shelf life of products [29,30]. Its role in probiotic technologies deserves special attention: for example, *Lactobacillus crispatus* produces PhLA along with lactic acid and hydrogen peroxide, which provides bactericidal activity against a wide range of pathogens, including antibiotic-resistant strains [31]. A number of studies have deciphered the mechanism of antimicrobial action of PhLA, which includes disruption of the integrity of the cell wall and membrane, interaction with genomic DNA and initiation of its degradation [24], suppression of the expression of virulence factors, which leads to damage to pathogen cells [28].

### 4.2. The Effect of Antibiotics on Metabolic Activity

The experiment examined the effects of 14 broad-spectrum antibiotics on the metabolic activity of hospital-acquired *K. pneumoniae* strains. Our findings reveal a heterogeneous response of *K. pneumoniae* to antimicrobial agents, underscoring that antibiotic exposure does not uniformly suppress bacterial metabolism. Three distinct patterns emerged.

Strong metabolic suppression: doxycycline, nitrofurantoin, and rifampicin consistently reduced PhLA concentrations across most strains. This suggests these drugs effectively inhibit central metabolic pathways or severely compromise bacterial viability, leading to a measurable decline in metabolic by-products such as PhLA. The uniformity of this response across strains may reflect the mode of action of these antibiotics, which target essential processes (e.g., protein synthesis, redox cycling) with broad metabolic consequences [32,33]. Metabolic maintenance or activation: meropenem, ciprofloxacin, ceftriaxone, cefepime, cefotaxime, cotrimoxazole, and ceftazidime either preserved PhLA levels near control values or led to elevated concentrations. This pattern implies that these antibiotics, while potentially bacteriostatic or bactericidal in standard susceptibility tests, do not fully arrest metabolic activity in the short term. Elevated PhLA in the presence of β-lactams (e.g., meropenem, ceftazidime) may indicate stress-induced metabolic reprogramming—for instance, activation of alternative pathways to counteract cell wall stress or oxidative damage [34,35,36]. Such responses could represent early adaptive mechanisms preceding phenotypic resistance. Variable/intermediate effects: clarithromycin and amikacin elicited mixed responses, ranging from suppression to activation of PhLA production. This strain-dependent variability may stem from differences in antibiotic uptake, efflux activity, or pre-existing metabolic states within the bacterial population. The inconsistent metabolic response to these agents highlights the complexity of predicting functional outcomes based solely on susceptibility categorization. Moreover, phenyllactic acid (PhLA) may exhibit direct antimicrobial activity against Klebsiella pneumoniae by disrupting cell wall membrane integrity and interfering with genomic DNA function [24].

Notably, in several cases, PhLA dynamics correlated with standard antibiotic susceptibility test results. Strains deemed susceptible by conventional methods often showed reduced PhLA levels when exposed to corresponding antibiotics (e.g., doxycycline, rifampicin), supporting the notion that metabolic suppression aligns with growth inhibition. Conversely, strains maintaining or increasing PhLA production under antibiotic pressure—even when classified as susceptible—may warrant closer scrutiny, as this could signal residual metabolic activity that might contribute to treatment persistence or relapse.

### 4.3. Quantitative Assessment of Microbial Cell and Phenyllactic Acid Levels

The characteristics of the PCR method highlight important aspects of the interpretation of the results. Since PCR determines the DNA content of microorganisms regardless of their viability, virtually no decrease in the number of *K. pneumoniae* was observed after incubation in the medium. However, we recorded an increase in the number of *K. pneumoniae* and the concentration of PLA for the G strain in the presence of amikacin and meropenem, and for the J strain in the presence of ceftazidime and clarithromycin. This is consistent with the results of a four-day experiment using doxycycline, in which it was found that, in addition to developing resistance, the microorganisms began to multiply faster. Colonies of mutated bacteria increased threefold compared to the baseline. Importantly, this effect was observed exclusively in bacteria exposed to the antibiotics and persisted even after the drug exposure was discontinued [37,38]. The study results suggest that antibiotic exposure triggers changes in the bacterial metabolome within the first minutes of exposure. Such modifications can initiate genetic disorders and contribute to the development of resistance [8,12,39,40]. This underscores the need for a comprehensive approach to assessing bacterial load.

### 4.4. Metabolic Activity Coefficient

To optimize data interpretation and visualize study results, a metabolic activity coefficient was developed. Graphical data representation using this indicator provides clarity of results, the ability to quickly analyze change dynamics, and facilitates comparison of indicators between different sample groups. Based on the metabolic activity coefficient, it was possible to clearly differentiate the inhibition of metabolic activity under the influence of doxycycline, nitrofurantoin, and rifampicin (with the exception of strain G). Remarkably, the ability to inhibit phenyllactic acid production was discovered in antibiotics that affect protein synthesis. Their mechanism of action is based on disrupting the functioning of bacterial ribosomes, which leads to the suppression of microbial growth or their death [41]. The recently discovered fact of ribosome heterogeneity may ensure rapid adaptation of bacteria to environmental stress factors [42]. Similar results were obtained in an experimental study with three clinically relevant species (*E. coli*, *S. aureus*, and *P. aeruginosa*). Among the seven classes of antibiotics studied, protein synthesis inhibitors produced the least heterogeneity in bacterial growth. However, the level of heterogeneity gradually increased with inhibitors of RNA synthesis, DNA replication, and drugs that disrupt cell membrane integrity and cell wall synthesis. From a clinical perspective, low heterogeneity is preferable, as high heterogeneity is often associated with bacterial survival during treatment [43]. This finding demonstrates additional aspects of the influence of antimicrobials on the metabolic profile of bacteria, expanding our understanding of their impact on the vital functions of pathogenic microorganisms.

Prospects for further research development include comprehensive work to improve the proposed method, including standardizing the methodology for calculating the metabolic activity coefficient, expanding the range of bacterial strains tested to obtain more representative data, validating the developed approach on clinical samples, and integrating the proposed method into existing microbiological testing protocols, thereby improving the efficiency of diagnosing and monitoring infectious diseases. The obtained results demonstrate the potential of the proposed method as an additional tool in the arsenal of modern antimicrobial therapy, capable of improving the treatment of infections caused by *K. pneumoniae*. The limitations of the method require further study of the mechanisms underlying the variability in the metabolic activity of different strains. It is particularly important to investigate cases of discrepancy between metabolic activity indices and the results of standard susceptibility tests.

Research shows that altering metabolism leads to the death of microbial cells, and activating immune cells through metabolic correction can improve the effectiveness of fighting infection [44]. Perhaps in the future, the development of analytical approaches in combination with mathematical methods will allow assessing changes in the metabolome and selecting an appropriate drug by calculation, preventing the selection of antibiotics and the development of resistance.

It should be noted that our work in this area is pilot and therefore has numerous limitations. Firstly, our study is based on a small data set, and therefore, all conclusions should be considered preliminary. We assessed changes in metabolic activity in vitro after incubation with only one metabolite, PhLA. We did not examine other bacterial species and are unable to assess changes in metabolic activity during microbial cooperation under real-life conditions; therefore, further research is needed. The precise mechanism of resistance development requires further investigation, as our current data do not permit a full assessment. Another limitation of this study is the use of RT-PCR, which allows for a quantitative assessment of the total bacterial load based on DNA but does not distinguish between viable and non-viable cells. For a more accurate determination, viability PCR (vPCR), which differentiates between viable microorganisms, should be used. Despite these limitations, we consider these preliminary findings important as they may lay the groundwork for a deeper understanding of the mechanisms of resistance in future studies. Further research incorporating larger datasets, multiple metabolites, diverse bacterial species, advanced molecular techniques, and in vivo models will be critical to validate and extend these initial observations.

## 5. Conclusions

In the study, phenyllactic acid (PhLA) was chosen as a marker of metabolic activity of *K. pneumoniae* since it was found to be the main metabolite of aromatic amino acids, accounting for more than 80% of the total. In vitro experiments have demonstrated the multidirectional effects of different groups of antibiotics on the metabolic activity of *K. pneumoniae.* Thus, we have shown that the presence of some antibiotics, such as doxycycline, nitrofurantoin, and rifampicin, significantly inhibits the metabolic activity of *K. pneumoniae* for most of the studied strains, resulting in a decrease in the concentration of PhLA in the medium. In the presence of meropenem, ciprofloxacin, cephalosporins (ceftriaxone, cefepime, cefotaxime, and ceftazidime), or cotrimoxazole, the concentration of PhLA in the medium was higher or similar to that in the control in most cases, indicating that the metabolic activity of *K. pneumoniae* strains remained high despite the presence of these antibiotics. In the case of clarithromycin and amikacin, different strains had different effects on the metabolism of *K. pneumoniae*, ranging from inhibition of PhLA production to activation of metabolic activity. The obtained results can serve as a basis for the development of innovative diagnostic tests and an in-depth study of the mechanisms of resistance formation in pathogenic microorganisms. In the future, the first results obtained in this study could help to limit the unnecessary use of certain antimicrobial drugs, which stimulate the metabolic activity of dangerous strains with multiple drug resistance. Of particular value is the fact that these patterns have not been previously described, which emphasizes the novelty and significance of the results obtained.

## Figures and Tables

**Figure 1 microorganisms-13-02599-f001:**
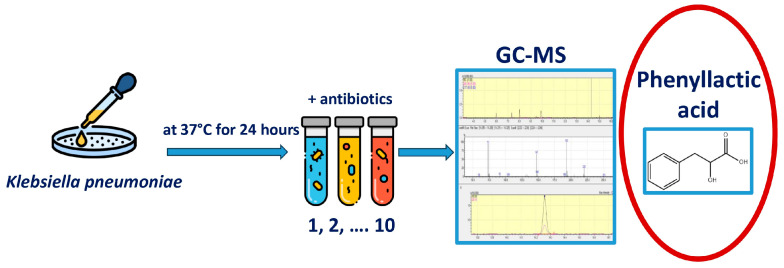
Experimental design. Primary seeding of clinical material was performed to isolate the *Klebsiella pneumoniae* strain. 1 mL of thioglycollate medium (TGM) was placed in Eppendorf tubes, 50 μL of a suspension of a daily *K. pneumoniae* culture was added, and a disk soaked in an antibiotic solution was placed and incubated at 37 °C for 24 h. After that, the concentrations of metabolites in the resulting suspension were determined.

**Figure 2 microorganisms-13-02599-f002:**
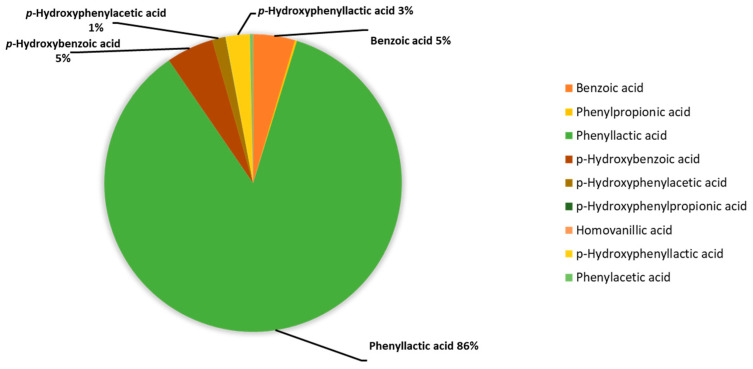
Percentage of phenylcarboxylic acids content in the assessment of the metabolic activity of *K. pneumoniae* strains determined using GC-MS, %.

**Figure 3 microorganisms-13-02599-f003:**
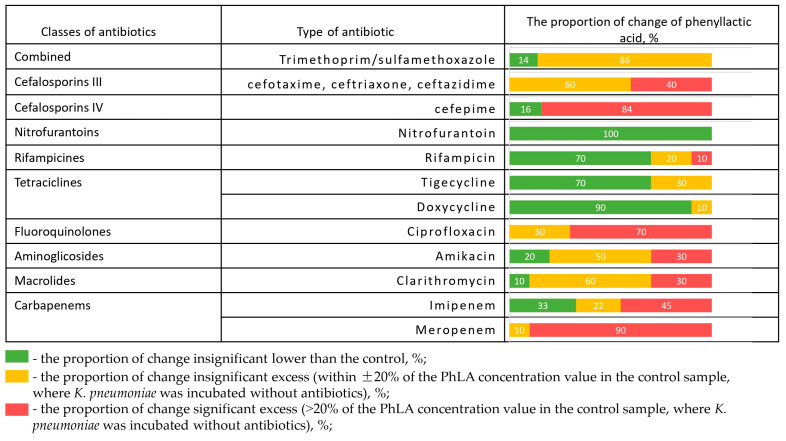
The proportion of change in the concentration of PhLA (%) upon administration of antibiotics relative to the control sample for hospital strains of *K. pneumoniae* (*n* = 10) after incubation in the presence of different antibiotics.

**Figure 4 microorganisms-13-02599-f004:**
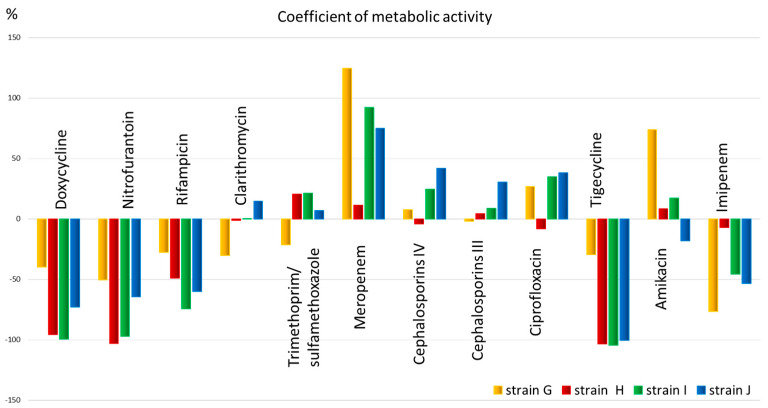
The metabolic activity coefficient for strains: G, H, I, and J, for 12 antibiotics corresponding to different classes.

**Table 1 microorganisms-13-02599-t001:** Hospital strains of *Klebsiella pneumoniae* and their characteristics.

Letter	Locus	Detectable Microorganisms
A	Broncho-alveolar lavage	*Klebsiella pneumoniae* 10^5^
B	Broncho-alveolar lavage	*Klebsiella pneumoniae* 10^7^*Acinetobacter baumanii* 10^7^
C	Punctuation	*Klebsiella pneumoniae* 10^4^
D	Broncho-alveolar lavage	*Klebsiella pneumoniae* 10^7^
E	Broncho-alveolar lavage	*Klebsiella pneumoniae* 10^4^*Acinetobacter baumanii* 10^4^
F	Wound or drainage	*Pseudomonas aeruginosa* 10^6^*Klebsiella pneumoniae* 10^6^
G	Whole blood	*Klebsiella pneumoniae,* carbapenemases of the groups were detected, OXA-48 and NDM
H	Contents of the abscess	*Klebsiella pneumoniae,* carbapenemases of the groups were detected, KPC и NDM
I	Whole blood	*Klebsiella pneumoniae,* carbapenemases of the groups were detected, KPC
J	Broncho-alveolar lavage	*Klebsiella pneumoniae* 10^4^

**Table 2 microorganisms-13-02599-t002:** The concentrations of antibiotics used in the experiment.

Name of the Drug	Concentration of Antibiotics (mkg)
Doxycycline	30
Nitrofurantoin	100
Rifampicin	5
Clarithromycin	15
Trimethoprim/sulfamethoxazole	1.25/23.75
Meropenem	10
Cefepime	30
Cefotaxime	5
Ciprofloxacin	5
Tigecycline	15
Amikacin	30
Imipenem	10

**Table 3 microorganisms-13-02599-t003:** The concentrations of metabolites obtained in the experiment, presented as the median for each acid in tubes with *K. pneumoniae* and antibiotics; and control tubes containing: thioglycollate medium; thioglycollate medium and 50 µL of *K. pneumoniae* microbial suspension; thioglycollate medium and a meropenem disk; thioglycollate medium and a doxycycline disk; thioglycollate medium and an amikacin disk; Mes (IQR25; IQR75) (*n* = 10), µmol/L.

Antibiotics		Metabolites, Mes (IQR25; IQR75) (*n* = 10), μmol/L							
	BA	PhPA	PhLA	*p*-HBA	*p*-HPhAA	*p*-HPhPA	HVA	*p*-HPhLA	PhAA
Doxycycline	10 (5.1; 11)	<0.5 (<0.5; <0.5)	2,2 (1.4;5.5)	1.4 (1.2; 1.8)	0.7 (0.6; 0.74)	<0.5 (<0.5; <0.5)	<0.5 (<0.5; <0.5)	0.6 (0.58; 0.9)	4. 5 (2.1; 6.4)
Nitrofurantoin	10 (5.2; 11)	<0.5 (<0.5; <0.5)	3.2 (1.9; 5.8)	1.4 (1.1; 1.9)	0.7 (0.6; 0.7)	<0.5 (<0.5; <0.5)	<0.5 (<0.5; <0.5)	0.7 (0.6; 0.8)	4.2 (2.6; 4,9)
Rifampicin	11 (5.5; 11)	<0.5 (<0.5; <0.5)	3.9 (2.4; 7.8)	1.5 (1.4; 1.6)	0.6 (0.6; 0.7)	<0.5 (<0.5; <0.5)	<0.5 (<0.5; <0.5)	0.7 (0.6; 0.8)	4.1 (1.9; 5.6)
Clarithromycin	11 (5.6; 11)	<0.5 (<0.5; <0.5)	8.8 (6.8; 11)	1.2 (1.1; 1.3)	0.7 (0.6; 0.7)	<0.5 (<0.5; <0.5)	<0.5 (<0.5; <0.5)	0.8 (0.7; 0.9)	4.1 (1.8; 5.1)
Trimethoprim/ sulfamethoxazole	11 (5.6; 11)	<0.5 (<0.5; <0.5)	8.5 (6.8; 11)	1.3 (1.2; 1.4)	0.7 (0.6; 0.7)	<0.5 (<0.5; <0.5)	<0.5 (<0.5; <0.5)	0.9 (0.7; 0.9)	4.2 (1.8; 4.6)
Meropenem	10 (5.5; 11)	<0.5 (<0.5; <0.5)	14.5 (8.5; 20)	1.3 (1.3; 1.4)	0.7 (0.6; 0.8)	<0.5 (<0.5; <0.5)	<0.5 (<0.5; <0.5)	1.0 (0.9; 1.2)	4.4 (1.9; 5.2)
Cephalosporins IV	10 (5.6; 11)	<0.5 (<0.5; <0.5)	12 (7.5; 13)	1.5 (1.2; 1.6)	0.7 (0.6; 0.8)	<0.5 (<0.5; <0.5)	<0.5 (<0.5; <0.5)	0.9 (0.8; 1.0)	4.4 (3.1; 5.4)
Cephalosporins III	10 (5.9; 12)	<0.5 (<0.5; <0.5)	10 (4.2; 12)	1.3 (1.2; 1.5)	0.7 (0.4; 0.8)	<0.5 (<0.5; <0.5)	<0.5 (<0.5; <0.5)	0.9 (0.7; 1.0)	4.2 (2.3; 4.7)
Ciprofloxacin	11 (5.2; 11)	<0.5 (<0.5; <0.5)	11 (6.9; 12)	1.3 (1.2; 1.5)	0.7 (0.5; 0.8)	<0.5 (<0.5; <0.5)	<0.5 (<0.5; <0.5)	0.9 (0.7; 1.2)	4.6 (1.8; 4.7)
Tigecycline	10 (5.6; 11)	<0.5 (<0.5; <0.5)	1.5 (1.1; 6.5)	1.2 (1.0; 1.4)	0.7 (0.4; 0.7)	<0.5 (<0.5; <0.5)	<0.5 (<0.5; <0.5)	0.6 (0.5; 0.7)	4.1 (1.9; 4.9)
Amikacin	11 (6.1; 11)	<0.5 (<0.5; <0.5)	8.8 (5.6; 11)	1.2 (1.1; 1.3)	0.7 (0.6; 0.7)	<0.5 (<0.5; <0.5)	<0.5 (<0.5; <0.5)	0.9 (0.7; 0.9)	4.6 (3.0; 4.7)
Imipenem	9.9 (5.0; 10)	<0.5 (<0.5; <0.5)	14 (4.1; 18)	1.3 (1.1; 1.3)	0.6 (0.5; 0.7)	<0.5 (<0.5; <0.5)	<0.5 (<0.5; <0.5)	1.0 (0.7; 1.2)	4.1 (2.0; 4.6)
TGM + *K. pneumonia*	11 (5.9; 12)	<0.5 (<0.5; <0.5)	8.8 (6.8; 13)	1.3 (11; 1.5)	0.7 (0.6; 0.8)	<0.5 (<0.5; <0.5)	<0.5 (<0.5; <0.5)	0.8 (0.7; 1.1)	4.5 (1.9; 5.3)
TGM	11 (5.0; 12)	<0.5 (<0.5; <0.5)	1.5 (1.3; 1.8)	0.9 (0.8; 1.0)	0.6 (0.4; 0.6)	<0.5 (<0.5; <0.5)	<0.5 (<0.5; <0.5)	0.6 (0.5; 0.7)	4.1 (1.4; 4.4)
TGM + Doxycycline	10 (4.1; 11)	<0.5 (<0.5; <0.5)	1.4 (1.1; 1.7)	0.9 (0.8; 1.0)	0.6 (0.4; 0.7)	<0.5 (<0.5; <0.5)	<0.5 (<0.5; <0.5)	0.6 (0.5; 0.7)	3.4 (1.7; 4.6)
TGM + Meropenem	10 (4.1; 12)	<0.5 (<0.5; <0.5)	1.5 (1.1; 1.7)	1.0 (0.9; 1.1)	0.7 (0.4; 0.7)	<0.5 (<0.5; <0.5)	<0.5 (<0.5; <0.5)	0.7 (0.5; 0.7)	3.4 (1.7; 5.2)
TGM + Amikacin	11 (3.9; 12)	<0.5 (<0.5; <0.5)	1.5 (0.9; 1.7)	1.0 (0.9; 1.0)	0.7 (0.3; 0.7)	<0.5 (<0.5; <0.5)	<0.5 (<0.5; <0.5)	0.6 (0.4; 0.6)	3.4 (1.6; 5.0)

**Table 4 microorganisms-13-02599-t004:** Determination of *K. pneumoniae* content by PCR after 24 h of exposure in the presence of antibiotics of different classes for strains G-J; 1 mL TGM + 50 μL *K. pneumoniae* + disk with antibiotic and concentration of PhLA, μmol/L in the corresponding samples.

Antibiotics	Strains							
	G		H		I		J	
	PCR, GE/mL	PhLA, μmol/L	PCR, GE/mL	PhLA, μmol/L	PCR, GE/mL	PhLA, μmol/L	PCR, GE/mL	PhLA, μmol/L
Doxycycline	8.0 × 10^6^	4.3	1.3 × 10^5^	0.0	1.2 × 10^6^	0.06	7.0 × 10^7^	2.4
Nitrofurantoin	2.0 × 10^7^	5.3	2.0 × 10^7^	0.2	1.2 × 10^6^	0.02	1.3 × 10^7^	1.9
Rifampicin	1.0 × 10^9^	6.4	1.0 × 10^8^	2.7	4.7 × 10^7^	0.59	2.0 × 10^7^	2.8
Clarithromycin	5.0 × 10^6^	6.1	4.0 × 10^6^	5.1	2.3 × 10^6^	2.3	2.0 × 10^7^	7.9
Trimethoprim/sulfamethoxazole	5.0 × 10^6^	6.9	3.0 × 10^6^	6.3	9.8 × 10^5^	2.8	7.4 × 10^6^	7.3
Meropenem	7.0 × 10^7^	19.7	4.0 × 10^6^	5.8	4.2 × 10^6^	4.5	8.0 × 10^6^	12.0
Cephalosporins IV	3.0 × 10^6^	9.5	7.0 × 10^6^	5.0	2.0 × 10^6^	2.9	4.0 × 10^6^	9.7
Cephalosporins III	1.2 × 10^6^	8.6	3.0 × 10^6^	5.4	1.6 × 10^6^	2.5	3.0 × 10^7^	9.0
Ciprofloxacin	2.6 × 10^6^	11.2	5.0 × 10^6^	4.8	7.3 × 10^5^	3.1	4.0 × 10^6^	9.5
Tigecycline	6.6 × 10^6^	6.2	7.0 × 10^4^	0.0	6.7 × 10^4^	0.0	6.0 × 10^6^	0.0
Amikacin	1.2 × 10^7^	15.3	5.0 × 10^6^	5.6	3.4 × 10^6^	2.7	2.4 × 10^7^	5.6
Imipenem	2.0 × 10^6^	2.1	2.0 × 10^6^	4.8	6.0 × 10^7^	1.3	1.4 × 10^5^	3.2
TGM + *K. pneumonia*	2.0 × 10^6^	8.8	2.0 × 10^6^	5.2	2.0 × 10^6^	2.3	5.0 × 10^6^	6.9
TGM	negative		1.4 × 10^4^		negative		negative	

## Data Availability

The original contributions presented in the study are included in the article/Appendix A, further inquiries can be directed to the corresponding author.The data presented in this study are available on request from the corresponding author due to (specify the reason for the restriction).

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
