# Peer review of "Phenyllactic Acid as a Marker of Antibiotic-Induced Metabolic Activity of Nosocomial Strains of Klebsiella pneumoniae In Vitro Experiment"

_microorganisms, 2025, doi:10.3390/microorganisms13112599_

Round 1

Reviewer 1 Report

Comments and Suggestions for Authors

as in vitro experiment  it is very difficult interpretate how this metabolic action  could be in vivo vivo. some sentences too long

need a table  or add to one of them results that compare control and add statistical results 

Klebsiella pneumoniae comments

Table 1.i   cursive letters ok, but correction write Klebsiella because all said Klebsiela

Table 2. Concentration of antibiotics mkg must change it use Eucast  or CLSI  instead those

Comments on the Quality of English Language

s in vitro experiment  it is very difficult interpretate how this metabolic action  could be in vivo vivo. some sentences too long

need a table  or add to one of them results that compare control and add statistical results 

Klebsiella pneumoniae comments

Table 1.i   cursive letters ok, but correction write Klebsiella because all said Klebsiela

Table 2. Concentration of antibiotics mkg must change it use Eucast  or CLSI  instead those

Author Response

Comment 1: as in vitro experiment  it is very difficult interpretate how this metabolic action  could be in vivo vivo. some sentences too long

Response 1: Thank you very much for your feedback and time. We fully agree that one of the limitations of the experiment is that we only analyzed one metabolite and cannot track all possible metabolic pathways and their changes. We have noted the limitations of our study and acknowledge that ours is a pilot study, and all conclusions require further clinical study and confirmation.

The text of the article has been significantly revised and we have tried to make the sentences shorter.

Comment 2: need a table  or add to one of them results that compare control and add statistical results 

Response 2: Thank you very much for your comment; it helped improve the presentation of our data and, we hope, made the article more understandable. We've added Table 3 with the results for all metabolites calculated as medians for all strains with added antibiotics, as well as in the control tubes.

Klebsiella pneumoniae comments

Comment 3: Table 1.i   cursive letters ok, but correction write Klebsiella because all said Klebsiela

Response 3: Thank you very much for the correction, it was a typo when entering the data into the table, we corrected it in table 1.

Comment 4: Table 2. Concentration of antibiotics mkg must change it use Eucast  or CLSI  instead those

Response 4: Thank you for your comment. We have rewritten the materials and methods section and described the use of commercial antibiotic susceptibility testing disks from BD (USA) and Bioanalyse (Turkey). Commercially produced disks contain the antibiotic concentration recommended for susceptibility testing according to EUCAST standards. The table lists the antibiotic concentrations taken from the instructions for the disks used.

Reviewer 2 Report

Comments and Suggestions for Authors

This manuscript analyzes the metabolic activity, specifically the production of phenylactic acid, of Klebsiella pneumoniae strains resistant to various antibiotics and isolated from a hospital setting. The manuscript presents several points that could be improved.

In the Materials and Methods section, section 2.1 on the preparation of the microbial suspension is very confusing; there is no order of events, and each point needs more precise detail. To improve it, it could be divided into different sections, starting, for example, with the origin of each isolate and then describing how their enrichment and susceptibility testing were performed. It also already mentions some of the results that should not be included in this section. The daily culture is not clear. Please explain what this refers to and what the growth conditions were.
The authors also mention that they included a tube with 1 ml of TGM as a control and a microbial suspension as a positive growth control. However, the negative control tube, which should include the culture medium and the antibiotic discs analyzed, or a disc with sterile paper, was missing. The DNA quantification section should also be separated into a separate subsection.

Section 2.2 also requires improved description: explain the internal controls used.

In the results section, it is not clear how the authors associate antimicrobial resistance to various antibiotics and their increase and decrease in PhLA production. They should specify how compensation for increased or decreased PhLA production influences microbial resistance capacity.
Failure to analyze all the detected metabolites and choosing only one creates bias, as some of the other metabolites may also influence bacterial resistance. It is not clear how PhLA production is associated with bacterial resistance. Please clarify or describe the possible mechanism.

The terminology is misused. The endpoint PCR technique cannot be used to correlate PHLA production, as it only allows us to detect total mass. A vPCR is required to determine viability.

The analysis shows no statistical correlation between resistance and PhLA production, so it cannot be suggested as a marker for this. The selected statistical tests are not suitable for analyzing the variables.

Author Response

This manuscript analyzes the metabolic activity, specifically the production of phenylactic acid, of Klebsiella pneumoniae strains resistant to various antibiotics and isolated from a hospital setting. The manuscript presents several points that could be improved.

Comment 1: In the Materials and Methods section, section 2.1 on the preparation of the microbial suspension is very confusing; there is no order of events, and each point needs more precise detail. To improve it, it could be divided into different sections, starting, for example, with the origin of each isolate and then describing how their enrichment and susceptibility testing were performed. It also already mentions some of the results that should not be included in this section. The daily culture is not clear. Please explain what this refers to and what the growth conditions were.
The authors also mention that they included a tube with 1 ml of TGM as a control and a microbial suspension as a positive growth control. However, the negative control tube, which should include the culture medium and the antibiotic discs analyzed, or a disc with sterile paper, was missing. The DNA quantification section should also be separated into a separate subsection.

Response 1: Thank you very much for your feedback. We've rewritten the materials and methods section. We've added a division into sections. We've tried to describe the experiment in more detail and clearly. We've added a description of the control tubes in the second stage of the experiment. We've also moved the description of DNA extraction into a separate section. We hope this section is now easier to understand.

Comment 2: Section 2.2 also requires improved description: explain the internal controls used.

Response 2: Clarifying text regarding internal controls has been added to the description of section 2.2.

For more accurate quantitative analysis using liquid-liquid extraction, surrogate internal standards not present in the test samples were used: D5-benzoic acid for benzoic acid conversion; 3,4-dihydroxybenzoic acid for phenylpropionic, phenylacetic, phenyllactic, p-hydroxyphenylpropionic, homovanillic, p-hydroxybenzoic, and p-hydroxyphenylacetic acids; and D3-hydroxyphenyllactic acid for p-hydroxyphenyllactic acid (Line 312-318).

Comment 3: In the results section, it is not clear how the authors associate antimicrobial resistance to various antibiotics and their increase and decrease in PhLA production. They should specify how compensation for increased or decreased PhLA production influences microbial resistance capacity.
Failure to analyze all the detected metabolites and choosing only one creates bias, as some of the other metabolites may also influence bacterial resistance. It is not clear how PhLA production is associated with bacterial resistance. Please clarify or describe the possible mechanism.

Response 3: Although changes in metabolic activity may indicate the development of resistance, further research aimed at elucidating the molecular mechanisms of resistance is required to confirm this hypothesis. A paragraph on possible resistance mechanisms has been added to the discussion in section 4.2., and the limitations have emphasized the preliminary nature of the interpretation of metabolic activity as a resistance marker.

Comment 4: The terminology is misused. The endpoint PCR technique cannot be used to correlate PHLA production, as it only allows us to detect total mass. A vPCR is required to determine viability.

Response 4: We agree that the use of PCR in our study does limit the ability to quantify microbial viability and, therefore, correlate with PHLA production. RT-PCR detects total DNA mass without distinguishing between live and dead cells, which explains the lack of a reliable correlation with PHLA concentration. This method was used to assess a possible increase in bacterial load. We recognize that methods such as vPCR are necessary for assessing viability and more accurate quantitative analysis, as discussed in the limitations section, and we will take this into account in future studies.

Comment 5: The analysis shows no statistical correlation between resistance and PhLA production, so it cannot be suggested as a marker for this. The selected statistical tests are not suitable for analyzing the variables.

Response 5: We fully agree with your comment, and the fragment has been removed from the article.

Reviewer 3 Report

Comments and Suggestions for Authors

Antibiotic-dependent changes in metabolic activity of nosocomial Klebsiella Pneumoniae strains assessed by Phenyllactic  Acid

There is a much scope of improvement in the manuscript before acceptance for publication.

  1. The title should be improved. It should include antibiotics. For eg; antibiotic-dependent changes in metabolic activity of nosocomial Klebsiella Pneumoniae strains assessed by Phenyllactic Acid (PhLA) for Resistance evaluation.
  2. The spelling of Klebsiella and Pseudomonas is misspelled throughout the manuscript in texts and tables. Hence, it is recommended to have your English checked by a professional writer.
  3. The introduction section should be improved. Pathogen significance, diagnostic challenges, metabolomic rationale, aim of the study.
  4. Instead of mentioning vague phrases like ‘standard test,’ it should specify standard susceptibility tests. Instead of ‘innovative diagnostic methods, mention a quantitative metabolic indicator. It provides a scientific and precise approach, drawing the attention of readers easily.
  5. The conclusion is based on a small data set; more bacterial strains should be tested, otherwise the conclusions should be considered preliminary.
  6. Only 1 metabolite was tested in vitro (PhLA). Metabolomic analysis is very small.
  7. The mechanism of resistance is also not fully assessed.
  8. Only in vitro findings are presented, which do not focus on its applicability.
  9. Should be improved, instead of labeling each slide. There should be a legend or colored boxes that match the slice color, along with the corresponding name and percentage of the compound. Like the green box, then Phenylacetic acid is 86%.
  10. The positioning of some citations needs to be changed. Instead, it should start at the end. Also, for multiple consecutive references, authors have used separate brackets, which need to be combined as on line 54. [9], [10], [11], [12] should be [9-12].
  11. Working in a hospital poses a risk of infection. However, the manuscript mentions that formal ethical approvals are not applicable. There should be a section in the manuscript that describes the details of the laboratory biosafety precautions taken by the authors.

Author Response

Antibiotic-dependent changes in metabolic activity of nosocomial Klebsiella Pneumoniae strains assessed by Phenyllactic  Acid

There is a much scope of improvement in the manuscript before acceptance for publication.

  1. The title should be improved. It should include antibiotics. For eg; antibiotic-dependent changes in metabolic activity of nosocomial Klebsiella Pneumoniae strains assessed by Phenyllactic Acid (PhLA) for Resistance evaluation.

Response 1: Thank you for your comment. We have taken into account the request to include a mention of antibiotics in the title and have proposed a new version: " Phenyllactic acid as a marker of antibiotic- induced metabolic activity of nosocomial strains of Klebsiella pneumoniae in vitro experiment." We are deliberately not mentioning the link to resistance, as at this stage of the study, a direct correlation between phenyllactic acid levels and antibiotic resistance has not yet been established experimentally. The current wording accurately reflects the subject of the study—the study of antibiotic-induced metabolic changes, with an emphasis on the role of PhLA as a potential marker.

  1. The spelling of Klebsiella and Pseudomonas is misspelled throughout the manuscript in texts and tables. Hence, it is recommended to have your English checked by a professional writer.

Response 2: Thank you very much for the correction, it was a typo when entering the data into the table, we corrected it in table 1.

  1. The introduction section should be improved. Pathogen significance, diagnostic challenges, metabolomic rationale, aim of the study.

Response 3:The introduction has been improved.

  1. Instead of mentioning vague phrases like ‘standard test,’ it should specify standard susceptibility tests. Instead of ‘innovative diagnostic methods, mention a quantitative metabolic indicator. It provides a scientific and precise approach, drawing the attention of readers easily.

Response 4: When revising this article, we made every effort to take your comments into account as much as possible and make the text more accessible.

  1. The conclusion is based on a small data set; more bacterial strains should be tested, otherwise the conclusions should be considered preliminary.

Response 5: Dear reviewer, we absolutely agree that this is a pilot study, as we indicate in the text of the article, and all conclusions should be considered preliminary.

  1. Only 1 metabolite was tested in vitro (PhLA). Metabolomic analysis is very small.

Response 6:The gas chromatography-mass spectrometry method we use involves the quantitative analysis of specific metabolites and differs in its capabilities from non-quantitative metabolomic analysis. In our study, we examined nine aromatic metabolites. Three of these are of particular interest due to their clinical significance demonstrated in previous studies. We refer to these metabolites as sepsis-associated metabolites and focus on them primarily: phenyllactic, hydroxyphenyllactic, and hydroxyphenylacetic acids. The primary goal of the experiment was to determine whether the concentration of these metabolites in K. pneumoniae changes with antibiotic exposure. We found that only the concentration of phenyllactic acid changes with antibiotic exposure. We therefore focused on this metabolite, recognizing that this approach is too narrow and will require further research, as we indicated in the limitations of our article.

  1. The mechanism of resistance is also not fully assessed.

Response 7: Although changes in metabolic activity may indicate the development of resistance, further research aimed at elucidating the molecular mechanisms of resistance is required to confirm this hypothesis. A paragraph on possible resistance mechanisms has been added to the relevant sections, and the limitations have emphasized the preliminary nature of the interpretation of metabolic activity as a resistance marker.

  1. Only in vitro findings are presented, which do not focus on its applicability.

Response 8: Thank you for your comments. We decided not to focus on the applicability of the method, as there is insufficient evidence to support the proposed hypotheses and the work requires further testing on a larger number of strains.

  1. Should be improved, instead of labeling each slide. There should be a legend or colored boxes that match the slice color, along with the corresponding name and percentage of the compound. Like the green box, then Phenylacetic acid is 86%.

Response 9: Thank you very much for your comment. We've modified Figures 2 and 3 and added a legend. Also, to improve understanding of the experiment, we've added Table 3 to the results section. It describes the results for all metabolites, calculated as medians for all strains with added antibiotics, as well as for the control tubes.

  1. The positioning of some citations needs to be changed. Instead, it should start at the end. Also, for multiple consecutive references, authors have used separate brackets, which need to be combined as on line 54. [9], [10], [11], [12] should be [9-12].

Response 10: Thank you, the corrections have been made.

  1. Working in a hospital poses a risk of infection. However, the manuscript mentions that formal ethical approvals are not applicable. There should be a section in the manuscript that describes the details of the laboratory biosafety precautions taken by the authors.

Response 11: While the study did not involve human subjects (hence, no ethical approval was required), we fully acknowledge the importance of biosafety in a hospital-associated laboratory setting. The work was conducted in a certified clinical microbiology laboratory, adhering to all relevant biosafety measures and laboratory control standards. These measures comply with international recommendations for biosafety in microbiology laboratories (including CDC and WHO guidelines and national standards), as well as the latest federal biosafety regulations of the Russian Federation. These protocols are aimed at minimizing the risk of occupational infection and are standard practice in clinical microbiology laboratories working with infectious agents.If necessary, we are prepared to include a detailed description of the protocols in the manuscript or attach relevant laboratory certificates and standards.

Round 2

Reviewer 1 Report

Comments and Suggestions for Authors

no further comments

Reviewer 2 Report

Comments and Suggestions for Authors

The authors have addressed the observations made. Thank you.

Reviewer 3 Report

Comments and Suggestions for Authors

authors has completed the suggested comment.